

# Inclusion of bedrock vadose zone in dynamic global vegetation models is key for simulating vegetation structure and functioning

Dana A Lapides[1], W Jesse Hahm[2], Matthew Forrest[3], Daniella M Rempe[4], Thomas Hickler[3], and David N Dralle[5]

[1]USDA Southwest Watershed Research Station, Tucson, AZ, USA
[2]Department of Geography, Simon Fraser University, Burnaby, BC, Canada
[3]Senckenberg Biodiversity and Climate Research Centre, Senckenberg, Germany
[4]Jackson School of Geosciences, University of Texas, Austin, Austin, TX, USA
[5]US Forest Service Pacific Southwest Research Station, Davis, CA, USA

**Correspondence:** Dana A Lapides (dana.lapides@usda.gov)

**Abstract.** Across many upland environments, soils are thin and plant roots extend into fractured and weathered bedrock where moisture and nutrients can be obtained. Root water extraction from unsaturated weathered bedrock is widespread and, in many environments, can explain gradients in vegetation community composition, transpiration, and plant sensitivity to climate. Despite increasing recognition of its importance, the "rock moisture" reservoir is rarely incorporated into vegetation and Earth system models. Here, we address this weakness in a widely used dynamic global vegetation model (DGVM, LPJ-GUESS). First, we use a water flux-tracking deficit approach to more accurately parameterize plant-accessible water storage capacity across the contiguous United States, which critically includes the water in bedrock below depths typically prescribed by soils databases. Secondly, we exploit field-based knowledge of contrasting plant-available water storage capacity in weathered bedrock across two bedrock types in the Northern California Coast Ranges as a detailed case-study. For the case study in Northern California, climate and soil water storage capacity are similar at the two study areas, but the site with thick weathered bedrock and ample rock moisture supports a mixed evergreen temperate broadleaf-needleleaf forest whereas the site with thin weathered bedrock and limited rock moisture supports an oak savanna. The distinct biomes, seasonality and magnitude of transpiration and primary productivity, and baseflow magnitudes only emerge from the DGVM when a new and simple subsurface storage structure and hydrology scheme is parameterized with storage capacities extending beyond the soil into the bedrock. Across the contiguous United States, the updated hydrology and subsurface storage improve annual evapotranspiration estimates as compared to satellite-derived products, particularly in seasonally dry regions. Specifically, the updated hydrology and subsurface storage allow for enhanced evapotranspiration through the dry season that better matches actual evapotranspiration patterns. While we made changes to both the subsurface water storage capacity and the hydrology, the most important impacts on model performance derive from changes to the subsurface water storage capacity. Our findings highlight the importance of rock moisture in explaining and predicting vegetation structure and function, particularly in seasonally dry climates. These findings motivate efforts to better incorporate the rock moisture reservoir into vegetation, climate, and landscape evolution models.



# 1 Introduction

Climate change is driving changes to precipitation, temperature, and fire regimes. Mediterranean areas in particular, which host significant amounts of the world's threatened plant biodiversity (Cowling et al., 1996), are projected to experience increased

precipitation volatility (Swain et al., 2018) and overall drier climate (Parmesan et al., 2022; Lee et al., 2021), including shorter wet seasons and an increase in the frequency and duration of conditions that result in extreme wildfire (Swain, 2021; Luković et al., 2021). To preserve these communities and better inform land management and climate adaptation research and policy, it is essential to understand how the current changes unfolding globally will impact plant communities in the decades to come. However, dynamic regional to global vegetation models–our most advanced tools for evaluating vegetation response to climatic

drivers–have historically struggled to capture vegetation behavior in seasonally dry environments, such as Mediterranean regions (Hickler et al., 2012, 2006).

Research insights from critical zone science may provide a clue as to why this may be the case. The critical life-supporting zone extends from the top of the vegetation canopy downward through typically thin, physically mobile soil and into deeper underlying saprolite and weathered bedrock layers (Anderson et al., 2004; Grant and Dietrich, 2017). Soil and bedrock in

the subsurface critical zone store and release water to plants and streams. Although extensive maps exist that can inform soil moisture properties in Earth system and vegetation models, mounting evidence suggests that i) many plants extensively exploit unsaturated moisture sourced from weathered bedrock below the mapped soil to sustain transpiration (McCormick et al., 2021), ii) the importance of shallow roots may be overestimated (Feddes et al., 2001), and iii) infiltrating rainfall and snowmelt in upland environments tend to transit this vadose zone rather than run off as Hortonion overland flow over the

surface (Salve et al., 2012; Hahm et al., 2022). Woody plant use of water stored beneath soils in weathered bedrock has been documented as early as the beginning of the 20th century (Cannon, 1911). 'Rock moisture', or water derived from the unsaturated weathered bedrock layer, is now understood to be an essential plant water reservoir, particularly in seasonally dry regions where it sustains transpiration later into the dry season (e.g., Schwinning, 2010; Rempe and Dietrich, 2018; Rose, 2003; McCormick et al., 2021; Hahm et al., 2022, 2020; Ruiz et al., 2010; Maysonnave et al., 2022). It has been difficult

to incorporate deeper water storage into dynamic vegetation models and Earth system models because weathered bedrock storage capacity has been historically challenging to quantify except at intensively monitored study sites. Recently, Wang-Erlandsson et al. (2016) presented a mass balance-based method for estimating total plant-available storage as the largest cumulative difference between incoming precipitation and outgoing evapotranspiration over a given time period, referred to as a 'deficit' (Grindley, 1960, 1968). For long time periods, the largest-observed deficit places a reasonable lower-bound on

the true plant-accessible water storage capacity, providing a method for estimating total plant-accessible water at large scales. McCormick et al. (2021) used a modification (Dralle et al., 2021) of this deficit-based approach to map rock moisture storage across the contiguous United States (CONUS) by subtracting soil water storage capacity (from Soil Survey Staff, 2019a) from the total plant-available storage capacity. They confirmed that large regions of the US Southwest and western Texas host





vegetation communities that rely on bedrock-derived water nearly every year. In such seasonally or intermittently dry areas, the subsurface is responsible for storing the water that supports plant communities through dry periods.

| Model name | Reference | Maximum soil depth | Number of layers | Variable soil depth? |
| --- | --- | --- | --- | --- |
| LPJ-GUESS | Smith et al. (2001), Smith et al. (2014) | 1.5 m | 2[‡] | N |
| IBIS | Pollard and Thompson (1995) | 4.25 m | 6 | N |
| MC1 | Daly et al. (2000) | 3 m | $\leq 10$ | Y |
| HYBRID | Friend et al. (1997) | N/A[**] | 1 | Y |
| SDGVM | Bond et al. (2005) | 1 m[†] | 4 | N |
| SEIB-DGVM | Sato et al. (2007) | 3 m | 3 | N |
| TRIFFID | Cox (2001) | 2 m[†] | 1 | N |
| CLM-DGVM | Lawrence et al. (2019) | Variable[*] | Variable | Y |
| ED, ED2 | Moorcroft et al. (2001) | Variable[*] | 1 | Y |

**Table 1.** Table of subsurface storage structures used in DGVMs. In the "Variable soil depth?" column, Y indicates that depth can vary by location, and N indicates that it does not vary by location. ‡ This study uses LPJ-GUESS version 4.0.1. The most recent version of LPJ-GUESS has 10 soil layers, although the soil depth is still 1.5 m. ∗∗ Soil water depth is not defined. Soil water capacity is defined. † This is the default value, but it may be possible to edit this in the model. ∗ Soil depth is determined from a soils dataset, which does not generally include weathered bedrock storage. Five of the nine DGVMs do not allow for soil depths greater than 3 m. Of the four models with spatially varying soil depth, the soil depth is specified from a soils dataset alone, which does not account for rock moisture.

Mediterranean regions where rock moisture appears to be important coincide with the regions where DGVMs and land surface models tend to underpredict dry-season plant transpiration and vegetation growth (e.g., Hickler et al., 2006, 2012), suggesting that incorporating rock moisture into DGVMs could improve performance in these regions. Across the set of widely used DGVMs included in Table 1, none explicitly account for rock moisture. Growing consensus in the hydrology community indicates that using soil properties to determine water availability to ecosystems neglects essential feedbacks between climate, ecosystems, and hydrology that determine subsurface water availability to plants (Gao et al., 2023). Further, subsurface plant water access (represented by rooting depth) has been demonstrated to be a strong control on DGVM results (e.g., Langan et al., 2017; Sakschewski et al., 2021). For these reasons, it may be important to improve prescriptions of the subsurface water storage accessible to plants, which can result in improved model performance (Piedallu et al., 2013). Jiménez-Rodríguez et al. (2022) implemented CLM-DGVM in Europe with two non-standard subsurface structures, both with soils 1.5 m deeper everywhere but using different soil textures for deeper 1.5 m soil layers. They found that the greater storage capacity allowed for better model performance in seasonally dry areas. This finding is suggestive, but without using realistic estimates of rock moisture storage, it is difficult to determine whether the model is 'getting the right answers for the right reasons' (Kirchner, 2006).

Here, we seek to investigate the impact of subsurface water flowpaths and deeper moisture supplies to plants by incorporating insights gained from an intensive field-based study of hillslope flow, water storage capacity and plant community composition



and function into the Lund-Potsdam-Jena GUESS Dynamic Global Vegetation Model (LPJ-GUESS DGVM; Smith et al., 2001, 2014). We alter model representation of subsurface structure to incorporate location-specific estimates of plant-accessible storage in weathered bedrock. Additionally, we develop a new hydrology scheme based on our best understanding of hydrological processes that results in increased infiltration into the vadose zone. Previous efforts focused on improving hydrological pro-
cesses in DGVMs have shown important model improvements arising from increased hydrologic realism, for example, by capturing the effects of topography and inter-pixel flow or improving inter-soil layer water transfer (Tang et al., 2014, 2015; Wolf et al., 2008a). We hypothesize that inclusion of plant-available water storage in weathered bedrock—in addition to soil water storage—in a DGVM would significantly improve the prediction of i) potential plant communities; ii) phenological patterns; and iii) summer dry season evapotranspiration (Gordon et al., 2004; Pappas et al., 2013; Eliades et al., 2018; Schwinning,
2010; Milly and Dunne, 1994; Pitman, 2003). We test this hypothesis in detail at two intensively monitored sites in Northern California with similar climate but distinct vegetation communities (Hahm et al., 2019) and more broadly at 4 km resolution across CONUS. This work provides a blueprint for incorporating deeper moisture stores into other DGVMs and Earth system models; accurately simulating this rock moisture reservoir is critical to understanding the impacts of global change on plants and water-carbon cycles in seasonally dry climates.

## 2    Methods

### 2.1    Field site descriptions

We build on recent studies (Hahm et al., 2019; Dralle et al., 2018, 2023b; Lovill et al., 2018) that found that within a large area of similar climate in the Northern California Coast Ranges, lithologically controlled differences in the extent of bedrock weathering and water storage capacity result in radical differences in plant communities (Figure 1) and their phenological
behavior as well as regional runoff patterns.

   Two intensively studied watersheds across a geologic contact reveal the role of bedrock water storage on plant water availability and streamflow generation. Elder Creek, a 16.9 km$^2$ watershed in the western Coastal Belt of the Franciscan Formation, receives around 2000 mm of annual precipitation, mostly as rain between November and April. The underlying bedrock consists primarily of shale (argillite) with some sandstone and conglomerate. The critical zone at Elder Creek has a thick unsaturated
zone and weathered, fractured bedrock (30 m thick at ridgetops) (Rempe and Dietrich, 2018; Salve et al., 2012), allowing ample water storage and supporting a dense evergreen forest canopy. Hydrological dynamics follow an annual cycle, with all rain infiltrating into the subsurface, increasing moisture in the soil and weathered bedrock vadose zone at the start of the wet season. The vadose zone recharges a hillslope groundwater aquifer (Dralle et al., 2023a), which flows laterally through fractures to generate streamflow, including both storm and baseflow (Dralle et al., 2018).
In contrast, Dry Creek is a smaller watershed (3.5 km$^2$) located about 20 km southeast of Elder Creek in the Central Belt of the Franciscan Formation. It receives approximately 1800 mm of annual precipitation. Dry Creek's lithology consists of mélange with intensely sheared, primarily argillaceous matrix (Hahm et al., 2019). The subsurface critical zone at Dry Creek is shallow, with thin organic soils and clay-rich weathered matrix overlaying unweathered, perennially saturated mélange found



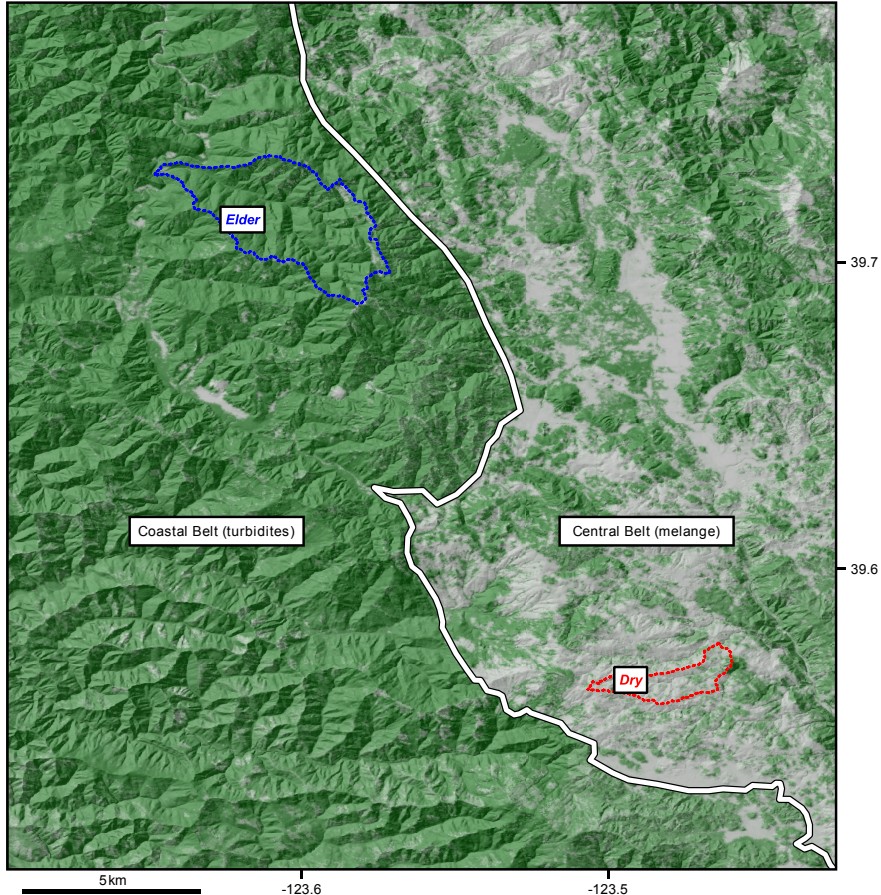

**Figure 1.** Map showing the locations of the two field study sites in Northern California, reprinted from Hahm et al. (2019). Elder Creek is positioned to the West of the geologic divide marked in white, while Dry Creek is positioned to the east so that the two sites have very similar climate but very different subsurface characteristics. Elder Creek has a large subsurface storage capacity, while Dry Creek has a small subsurface storage capacity. Coordinates reported in WGS84, geologic contact after (Jayko et al., 1989), canopy cover from the 2011 National Land Cover Database.

just 2-4 m below the surface (Hahm et al., 2020). As a result of limited weathering, Dry Creek has limited subsurface water storage capacity, which results in a winter-deciduous oak annual grassland savanna as the primary vegetation community, despite similar rainfall totals to Elder Creek. Across the state of California, the Central Belt mélange has a lower plant-water storage capacity than other rock types in a similar climate (Hahm et al., 2023).

5  Thus, the differing lithologies and critical zone structures between the two sites lead to significant disparities in storage dynamics and vegetation communities. Elder Creek's thick subsurface critical zone supports a dense evergreen mixed broadleaf-needleleaf forest canopy, while Dry Creek's shallow subsurface critical zone can only sustain a grassland savanna.







**Figure 2.** Schematic representing (left) the default LPJ-GUESS soil storage hydrology vs. (right) the modified LPJ-GUESS soil storage hydrology. See Section 2.2 for more details. Roman numerals indicate the order in which processes are simulated. PFT is plant functional type, 'T' transpiration, 'E' evaporation, 'rain_and_snowmelt' the combination of input rain water and snowmelt, $Q_{surf}$ surface runoff, $Q_{percolate}$ percolation of water from upper soil layer to lower soil layer, $Q_{drain}$ a lateral drainage term of water leaving the lower soil layer, $Q_{baseflow}$ a runoff term leaving the lower soil layer, $wcont_0$ and $wcont_1$ the water content of the upper and lower soil layers, $\theta_0$ and $\theta_1$ the volumetric water content of the upper and lower soil layers, and $\theta_{0,max}$ and $\theta_{1,max}$ the volumetric water content at field capacity for the upper and lower soil layers, respectively.

## 2.2 Model description

LPJ-GUESS is a process-based dynamic regional to global vegetation model that represents plant physiological and bio-geochemical processes as well as detailed representations of tree population dynamics, disturbance by wildfires and biome biogeography (Sitch et al., 2003; Smith et al., 2001). LPJ-GUESS has been successfully used for a variety of applications in



| Parameter/process | Default | Modified | Justification | Effect |
|---|---|---|---|---|
| Subsurface storage capacity | Two layers of depths 50 cm and 100 cm. | Two layers with depths defined by a soil storage capacity and a rock moisture storage capacity, respectively. | Widespread evidence that plants access water from weathered bedrock underlying soil result in the need to make location-specific root-zone storage capacities that can be much greater than or less than the default capacity. | Places with larger storage capacity can support transpiration longer through dry periods. |
| Soil water transport capacity | Limited by a slow drainage capacity (maximum ≈2 mm/day. | Limited by $K_{sat}$. | $K_{sat}$ is the measured transport capacity of soils at saturation, which can be many orders of magnitude greater than the 2 mm/day prescribed by the default model | More water infiltrates into lower soil layer. |
| Surface flow | Overflow from upper soil when at field capacity. | Rainfall in excess of $K_{sat}$. Overflow when at field capacity is routed to deeper soil/weathered bedrock layer. | Horton overland flow occurs when infiltration capacity (at least $K_{sat}$) is slower than the rainfall rate. To model saturation-excess overland flow (SOF), knowledge of confining layers is necessary. Instead we group SOF in with all other subsurface flows. | More water infiltrates into the lower soil layer. Surface flow is no longer the primary runoff mechanism. |
| Subsurface flow | $Q_{drain}$ is runoff from lower soil layer in excess of field capacity. $Q_{baseflow}$ is a slow drainage term from the lower soil layer. | All water from the lower soil/weathered bedrock layer in excess of field capacity is combined into one $Q_{baseflow}$ term. | More detailed process representation is required to distinguish between runoff generation mechanisms. | All subsurface-mediated runoff is combined into a single term, which becomes the most important runoff mechanism. |

**Table 2.** Table summarizing modifications made to the LPJ-GUESS hydrology scheme.





vegetation change, carbon cycling, and biomass modeling (e.g., Hickler et al., 2004; Steinkamp and Hickler, 2015; Wolf et al., 2008b; Hickler et al., 2012; Miller and Smith, 2012; Cai et al., 2022; Badeck et al., 2001; Morales et al., 2005; Bondeau et al., 2007; Yurova and Lankreijer, 2007). A full list of publications using LPJ-GUESS can be found at https://web.nateko.lu.se/lpj-guess/index.html. A full description of the LPJ-GUESS model can be found in Smith et al. (2001) and

Smith et al. (2014), with the default hydrology scheme described by Gerten et al. (2004). In this study, we work from LPJ-GUESS version 4.0.1 (https://web.nateko.lu.se/lpj-guess/index.html). More recent version of LPJ-GUESS have increased vertical soil layer resolution to better resolve soil temperature. We made two distinct updates to the subsurface hydrology scheme in LPJ-GUESS. The first update is a change to the subsurface storage capacity, and the second is a change to the hydrological processes. We refer to these throughout as the "modified" storage and hydrology, respectively, as compared to the "default"

storage and hydrology. Figure 2 shows a schematic diagram of the subsurface hydrology for the default storage capacity and hydrology (left) and the modified storage capacity and hydrology (right). Here, we discuss the differences between the two schemes, summarized in Table 2.

### 2.2.1 Changes to subsurface storage capacity

In LPJ-GUESSv4.01, the plant-available water storage capacity is divided between an upper and a lower subsurface layer. In

the default storage capacity, the two soil layer thicknesses are globally uniform (0.5 and 1 m for the upper and lower layers, respectively). The storage capacity varies by location only due to variability in soil properties that alter the water-holding capacity of the substrate. However, the size of this subsurface storage reservoir is crucial to plant functioning through dry periods. If geology permits, and climate supplies enough water, plants can expand their root systems to access water stored outside of this 1.5 m zone. In order to capture plant functioning through dry periods, it is essential to incorporate a more

accurately sized storage reservoir. Thus, in the modified storage capacity, the upper soil layer depth is defined by a soil capacity specified from a soils dataset, and the lower layer depth is defined by a storage capacity specified from a rock moisture dataset. In general, this difference from the default layer depths will result in greater root zone depth in the modified storage capacity, but the magnitude and direction of change is location-specific. It is important that the rock moisture be location-specific rather than quasi-unlimited everywhere since the latter formulation can result in a model with no water limitation, which is not

realistic. By setting a location-specific depth, water stress is allowed but only when a realistic storage capacity is depleted.

### 2.2.2 Changes to flow processes

Each gridcell in LPJ-GUESS functions as a separate 1–D column (i.e., there is no lateral flow). The input flux at the top of the column is from rain or snowmelt, which recharges the upper layer. Abiotic evaporation can remove water from the uppermost portion of the the upper layer, until the wilting point is reached. Transpiration can remove water from throughout both the

upper and lower layer, based on the availability of water in these layers, leaf and atmospheric demand, and root distribution of present PFTs. The details of the hydrologic processes determine how much water enters and leaves each soil layer, and where it goes when it leaves. There are three main differences between the default and modified hydrology introduced here, which



pertain to: (i) overland flow runoff generation, (ii) percolation from the upper to the lower soil (or weathered bedrock) layer, and (iii) runoff generation from the lower soil (or weathered bedrock) layer.

In the default hydrology, overland flow ($Q_{surf}$) can remove water from the upper layer only when field capacity is exceeded. In the modified hydrology, we instead allow for $Q_{surf}$ when the intensity of rainfall exceeds the infiltration capacity of the

soil ($K_{sat}$), which explicitly represents the process of infiltration excess or Horton overland flow (HOF, Horton, 1933, 1945). When field capacity is reached, excess water percolates (at a rate not exceeding the saturated hydraulic conductivity, $K_{sat}$) to the lower layer rather than leaving the soil column as surface runoff. The overland flow mechanism of saturation overland flow (SOF, Dunne and Black, 1970) occurs when the subsurface is fully saturated in a manner similar to $Q_{surf}$ in the default hydrology. However, without a confining layer, this mechanism would require saturation of both soil layers (rather than just the

top layer) before producing surface flows. Without additional process modeling and details on subsurface structure, it is most reasonable to assume that water would infiltrate to the lower soil layer before producing runoff. If SOF did occur once both layers were saturated, it would be classified as subsurface-mediated flow in the modified hydrology, leaving from the lower layer. However, this classification is reasonable for SOF, which does involve significant mixing with subsurface waters prior to producing flows (Lapides et al., 2022).

In the default hydrology, percolation from the upper layer to the lower layer and to baseflow from the lower soil layer occurs at a percolation rate which is determined from soil textural properties and multiplied by the square of the relative plant-available water content (PAWC) in the layer. This percolation rate, when at field capacity, is at most ≈2 mm/day. Importantly, the slow rate of percolation from the upper soil layer in the default hydrology limits the amount of water that can infiltrate into the lower soil layer so that, once the upper soil layer reaches field capacity, nearly all rainfall is transported out of the column as $Q_{surf}$. In

the modified hydrology, we allow percolation from the upper to lower layer at a rate of $K_{sat}$. For comparison, $K_{sat}$ in forested areas can be as high as 10s or 100s of mm/hour (e.g., Godsey and Elsenbeer, 2002; Elsenbeer et al., 1999; Davis et al., 1996), orders of magnitude larger than the default percolation rate. Thus, in the modified model, more water is transported into storage in the lower layer before runoff is generated.

While the default hydrology provides two pathways for runoff from the lower layer, the modified hydrology has only one.

The slow drainage term called $Q_{baseflow}$ and the $Q_{drain}$ runoff term in the default hydrology are replaced in the modified hydrology by a single baseflow term $Q_{baseflow}$ that delivers all water above field capacity out of the column. In effect, the modified hydrology $Q_{baseflow}$ is the same as the default hydrology $Q_{drain}$, and the default $Q_{baseflow}$ is removed from the modified hydrology. This means that when the lower layers is below field capacity in the modified hydrology, there is no runoff or drainage (compared to the slow drip of $Q_{baseflow}$ in the default model), retaining more water in the lower layer for plant use.

Overall, the processes in the default hydrology can be interpreted as quickflow that relatively rapidly flows to streams ($Q_{surf}$ and $Q_{drain}$) and a baseflow term ($Q_{baseflow}$) that recharges groundwater that more slowly makes its way to streams. In the modified hydrology, the processes can be interpreted as Horton overland flow ($Q_{surf}$) and groundwater-mediated runoff ($Q_{baseflow}$). Groundwater-mediated runoff conceptually includes: saturation overland flow (Dunne and Black, 1970) which is a groundwater-mediated flow (Lapides et al., 2022), drainage to groundwater, and any lateral flows out of the column. Without

simulating groundwater levels or routing flows, this 1-dimensional hydrology scheme allows us to simply account for runoff in



a mass-balance sense, although we do not know specifically which groundwater-mediated runoff mechanisms are at play in a given location without a more detailed modeling approach. Both the upper and lower layers in the modified hydrology function with a threshold drainage behavior that matches contemporary understanding of runoff generation on hillslopes (Spence, 2010).

### 2.2.3 Effects of the modified hydrology scheme

5 The change to realistic soil and weathered bedrock storage capacities from a globally uniform soil depth generally increases water storage capacity. This increase in capacity provides more space to store plant-available water that is accessible during dry periods. The changes to the hydrology scheme further act to enhance subsurface water availability to plants. First, more water percolates from the upper soil layer to the lower layer due to the increased soil transport capacity in the modified model, retaining more water in the root zone rather than immediately generating surface runoff. Second, runoff is not generated from

10 the lower layer in the modified hydrology unless it has reached field capacity, so all unsaturated moisture below field capacity (and above wilting point) is retained for plant water use in the lower layer. The other major effect of the modified hydrology scheme is that $Q_{surf}$ becomes a negligible term, and runoff is predominantly made up of $Q_{baseflow}$, which reflects better the understanding that most runoff is generated via groundwater-mediated mechanisms rather than as overland flow, particularly in Mediterranean environments(Salve et al., 2012; Hahm et al., 2022).

15 ## 2.3 Data sources

| Data type | Data source | Native resolution | Use |
|---|---|---|---|
| Precipitation | PRISM[1] | 4 km | Input and Evaluation |
| Air temperature | PRISM[1] | 4 km | Input |
| Shortwave radiation | Daymet[2] | 1 km | Input |
| $CO_2$ concentrations | ACCMIP, as processed by (3) | 0.5° | Input |
| N concentrations | ACCMIP, as processed by (3) | 0.5° | Input |
| Soil texture | LPJ-GUESS[3] | 0.5° | Input |
| Soil depth | gNATSGO4 via (5) | 500 m | Input |
| Rock moisture | Derived from (5) | 500 m | Input |
| ET | PML-V2[6] | 500 m | Evaluation |
| LAI | MODIS LAI[7] | 500 m | Evaluation |
| Land cover | NLCD[8] | Processing | |
| GPP | Benchmark[9] | Evaluation | |

**Table 3.** Distributed data sources used for input to LPJ-GUESS and evaluation of results. 1 - PRISM Climate Group (2014), 2 - Thornton et al. (2022), 3 - Smith et al. (2014),4 - Soil Survey Staff (2019b), 5 - McCormick et al. (2021), 6 - Zhang et al. (2019), Gan et al. (2018), and Zhang et al. (2016), 7 - Wang et al. (2022), 8 - Jin et al. (2023), 9 - Seiler et al. (2022).




Historical climate data for the period from 1981-2021 were compiled from the sources listed in Table 3. All data were regridded to match the 4 km grid used for the PRISM dataset since that is the lowest-resolution forcing data source. PRISM precipitation and PML-V2 have been found to perform well for mass balance closure compared with USGS streamflow gages (Rempe et al., 2023). For case study model runs, soil depth and rock moisture were derived from field-based estimates (Rempe

and Dietrich, 2018; Dralle et al., 2018; Hahm et al., 2019, 2020). For CONUS model runs, soil depth and rock moisture are derived from the datasets available through Dralle et al. (2021), including the downsampled gNATSGO (Soil Survey Staff, 2019b) and rock moisture storage. We masked out areas where evapotranspiration (ET) exceeds precipitation (P) over the period 2003-2017 and areas with negative estimated rock moisture storage. These criteria help to ensure that ET is not supplemented significantly by irrigation and that pixels with negative rock moisture storage estimates are not fed into LPJ-GUESS. We then

converted the rock moisture storage capacity to a depth for the lower layer using the soil texture characteristics, as specified by Sitch et al. (2003). Since plant water access is not restricted by depth within the second soil layer, it is important only that the storage capacity of the lower layer is reflective of natural conditions, not its depth. For this reason, it is convenient to use the same soil texture for both layers since depth can be tuned to achieve the correct storage. Prior to model evaluation, we masked out pixels classified as open water, developed, or cultivated land cover types in the National Land Cover Database (Jin et al.,

15 2023).

## 2.4 Model runs

We ran LPJ-GUESS for case study sites at Elder Creek and Dry Creek and then across CONUS. For all model runs, the nitrogen cycle was enabled, and land use was not included, so simulation results represent potential natural vegetation. For all locations, we ran four different simulations based on the same climate data. These simulations include each combination of the default

storage capacity and hydrology and the modified storage capacity and hydrology: (i) default storage capacity and hydrology, (ii) modified storage capacity with default hydrology, (iii) default storage capacity with modified hydrology, and (iv) modified storage capacity and hydrology, as shown in Figure 3. For the case study locations, climate is nearly the same at both sites, and the soil texture is the same. This means that the model runs with default storage capacity (i, iii) are essentially identical for the two sites, so only a single output is shown to represent these cases in the case study results.

## 25 2.5 Model evaluation

Case study results for Elder and Dry Creek were evaluated based on field observations of plant communities. All output was evaluated based on comparison between mean annual and mean summer (July-September) ET signatures produced by LPJ-GUESS and PML-V2 (Zhang et al., 2019). Pixel-wise annual runoff was estimated as mean annual evapotranspiration from PML-V2 subtracted from mean annual precipitation from PRISM. This estimated annual runoff was also used for validation

purposes, given the introduction of a new hydrology scheme (See Figure S1). Since there is no saturated zone model, stream routing algorithm, or lateral flow in LPJ-GUESS, the absolute timing of runoff cannot be compared directly to hydrograph data, but integrated seasonal and annual runoff totals should approximate basin-scale runoff, which has been proven to be an appropriate approach at the coarse scale at which the model often is applied (0.5 degrees, i.e. roughly 50 km pixels; Gerten







**Figure 3.** Key to describe the four model structures used in this study as combinations of two hydrology schemes (rows) and two storage structures (columns). Colors correspond to the colors used to represent each model structure in Section **??**. More details about the model structures can be found in Section 2.2.

et al., 2004). When comparing LPJ-GUESS modeled ET to PML-V2, output data were restricted to the period 2000-2021 for which PML-V2 is available.

For the case study sites, vegetation community composition was assessed using MODIS LAI (Myneni et al., 2021), with a mean and standard deviation LAI of $4.4 \pm 0.85$ for Elder Creek and $1.6 \pm 0.19$ for Dry Creek. Based on field observations, we

5   estimated a fraction of LAI expected for trees vs grass at each site. For Elder Creek, given nearly full forest cover, we expect that 75-100% of LAI is accounted for by trees. At Dry Creek, given the oak woodland structure, we expected that 10-50% of LAI is accounted for by trees.

We used annual runoff from mass balance, annual ET from PML-V2, and summer ET from PML-V2 to evaluate model performance based on Kling-Gupta Efficiency (KGE; Gupta et al., 2009) and a spatial distribution metric (Seiler et al., 2022).

10   Model performance results are summarized in Supplemental Information S1. Overall, annual runoff and ET performance is slightly decreased with the modified model but still good, and summer ET performance is substantially increased.





**Figure 4.** Monthly mean relative water content in the (a) upper soil layer, (b) lower soil or weathered bedrock layer and (c) total root zone for each model setup. Squares denote the modified hydrology scheme, and circles denote the default hydrology scheme. Color indicates the subsurface storage, with Blue default, green Dry Creek, and red Elder Creek. Default and Dry Creek water contents go to zero in the summer for both hydrology schemes. Only the large storage capacity for Elder Creek allows for sustained water supply through the summer. By comparing the circles and squares for each storage, it is clear that more water enters the lower layer with the modified hydrology scheme in the wet season, leading to enhanced water availability even with the same storage.





**Figure 5.** Average maximum annual moisture storage in each soil or weathered bedrock layer and across the full root zone for all four simulations (a) across CONUS and (b) for pixels where storage capacity increases by more than 200 mm. Moving from the default to modified hydrology increases available moisture (see difference between blue x and green square and between pink triangle and orange circle). Moving from default to modified storage capacity substantially increases available moisture in general (compare blue x and green square to pink triangle and orange circle). Differences between simulations are more substantial with larger storage increase in panel b. (c) Minimum root zone storage as a function of difference in storage capacity from the default. With large storage capacity, significant amounts of water remain in storage at all times. (d) Difference in maximum mean monthly plant-available moisture from the default model to each simulation. As the difference in storage capacity grows, up to an average of 300 mm more moisture is available. In both panels, error bars show the range from 25-75th percentile.





## 3 Results

### 3.1 Modified hydrology increases plant-available storage

The modified hydrology scheme increases infiltration into the lower soil/weathered bedrock layer. This is clear to see in the case study results (compare squares to circles for each color in Figure 4b). The increase in available water becomes larger as the size of the lower layer increases from Dry Creek (green) to Default (blue) to Elder creek (red). The observation that (with the modified hydrology) all curves meet the maximum relative water content at the end of the wet season in March in Figure 4c indicates that the climate is able to supply water to fill larger subsurface storage. Thus, at Elder Creek, increasing total root zone storage capacity substantially increases plant-available water, especially through the summer as water use draws down storage.

These same observations hold across the contiguous United States (CONUS); modified hydrology and storage capacity generally result in more plant-available moisture (Figure 5a and b). For locations where storage capacity increases by more than 200 mm between the default and modified storage capacity, differences between simulations in plant-available moisture are magnified (Figure 5b). At an annual level, modified hydrology increases root-zone storage, but modified storage capacity has an even bigger impact on root-zone storage. As the modified storage capacity increases, maximal plant-available moisture with the modified storage capacity and hydrology grows up to a median of about 300 mm more than with the defaults (Figure 5d).

### 3.2 More plant-accessible water results in more transpiration

Enhanced water availability translates into overall greater plant transpiration at the case study sites (T, Figure 6a-b). Switching to the modified hydrology scheme (squares) results in an enhanced T curve with a similar seasonal pattern. As storage capacity increases from Dry Creek (green) to Default (blue) to Elder Creek (red), T shifts later into the summer, better matching T derived from the satellite-derived product PML-V2 (Zhang et al., 2019).

With modified storage capacity and hydrology, T is enhanced across CONUS relative to the default model (Figure 7a-c, Figure 8a), with a median increase of 100-150 mm annually. This effect is particularly notable in the late summer months, when monthly T can increase by a median of 20 mm (Figure 8b-d). As with the case study, both modified hydrology (Figure 7b) and modified storage capacity (Figure 7c) result in generally greater T, with an additive effect between both changes resulting in strong increases in ET along the West Coast, Texas, and the Southeast (Figure 7a). The regions with greatest increase in ET match with the areas with highest storage capacity in the modified storage capacity model (Figure 7d). The effect is strongest in the intersection between areas with large root-zone storage capacity and high asynchronicity index (Figure 7e). T increases are largely limited to temperate dry summer and temperate no-dry season climates (Figure 7f).

To compare with the satellite-derived product PML-V2 product, we use the modeled ET, which includes, in addition to T, modeled soil evaporation and interception. While we generally report changes in T to highlight the plant response, we compare ET from LPJ-GUESS to PML-V2 to ensure that different partitioning between evaporation and T does not affect the comparison. The increase in summer T results in more accurate summer ET (July-September) across CONUS, when compared



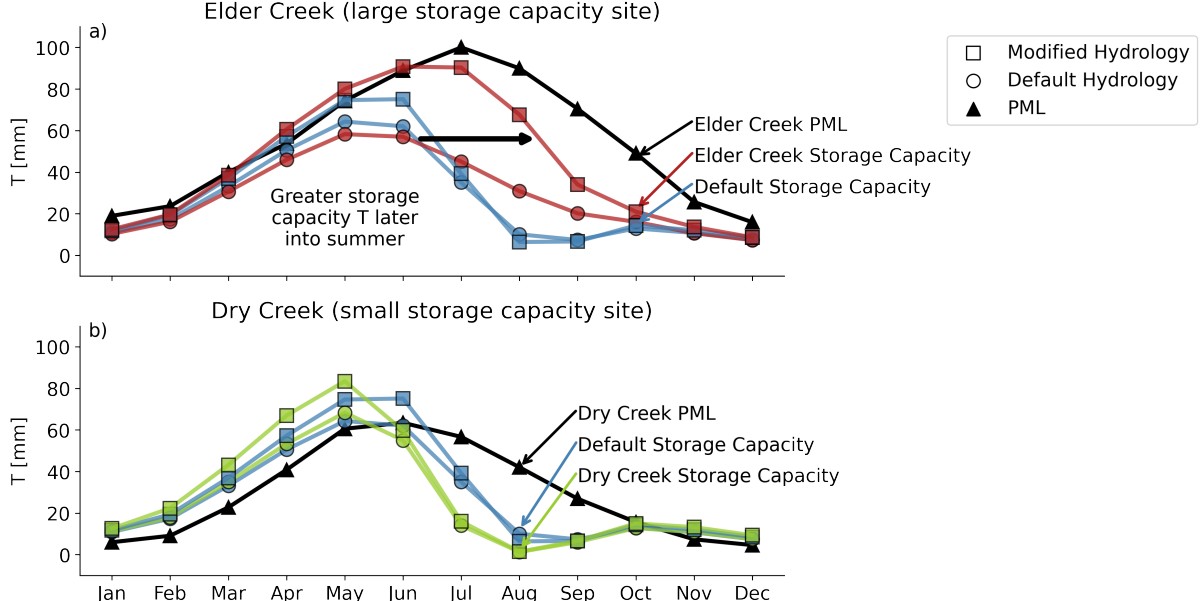

**Figure 6.** Comparison between average monthly ET as estimated by LPJ-GUESS using different storage and hydrology and from PML-V2 for (a) Elder Creek, which has large subsurface storage capacity and (b) Dry Creek, which has small subsurface storage capacity. Transpiration from PML (black triangles) shows sustained T into the dry season. LPJ-GUESS best approximates this behavior at Elder Creek with both greater subsurface storage capacity and the modified hydrology routine.

to PML-V2 (Figure 9a). KGE between mean annual summer ET from LPJ-GUESS and PML-V2 improves from 0.27 for the standard model to 0.89 with the modified model (see Supplemental Information for more details). As the difference in storage capacity between the modified and default models increases, the fit between PML-V2 and the default LPJ-GUESS summer ET becomes worse, with the highest median error around 100 mm from July-September (Figure 9a). With modified storage capacity and hydrology, fit is much better, median errors nearly vanish until the largest storage capacity bins (Figure 9a). Across CONUS, these error reductions are strongest in the west and western Texas, although improvements are visible across most of the area (Figure 9c-d).

### 3.3 Differences in T correspond to changes in plant community

In the case studies, differences in T translate to distinct plant communities predicted for each model scenario. This can be represented as a position in a 2-D space relating the LAI of grasses and trees (Figure 10). Shaded regions denote the estimated space in which plant community should fall for (red) Elder Creek and (green) Dry Creek, based on MODIS LAI and field expertise. Using the default storage capacity, LPJ-GUESS predicts the same plant community for Elder and Dry Creek (blue) with essentially all trees with the default hydrology (blue circle) and about 80% trees with the modified hydrology (blue square). With the modified storage capacity but the default hydrology, Dry Creek (green circle) and Elder Creek (red circle) are



**Figure 7.** (a)-(c) Difference in simulated annual T between each modified simulation and the simulation with default storage capacity and hydrology. In (a), regions with largest enhancement in T from modifications to storage and hydrology include the West Coast, Texas, and the Southeast. (d) Root-zone storage used for the modified storage capacity, as described in Section 2.3. Largest storage capacities are found in California and in a vertical band running north from Texas. (e) Asynchronicity index, demonstrating difference in seasonality between ET and precipitation at each pixel, following Feng et al. (2019). See Section 2.3 for more detail. (f) Koppen Geiger Climate Classification from Kottek et al. (2006). Symbols are defined as: Af–tropical rainforest, Am–Tropical monsoon, Aw–Tropical savanna, BS–Dry Semi-Arid, BW–Dry Arid Desert, Cf–Temperate No dry season, Cs–Temperate Dry summer, Df–Continental No dry season, Ds–Continental Dry summer, ET–Polar Tundra.



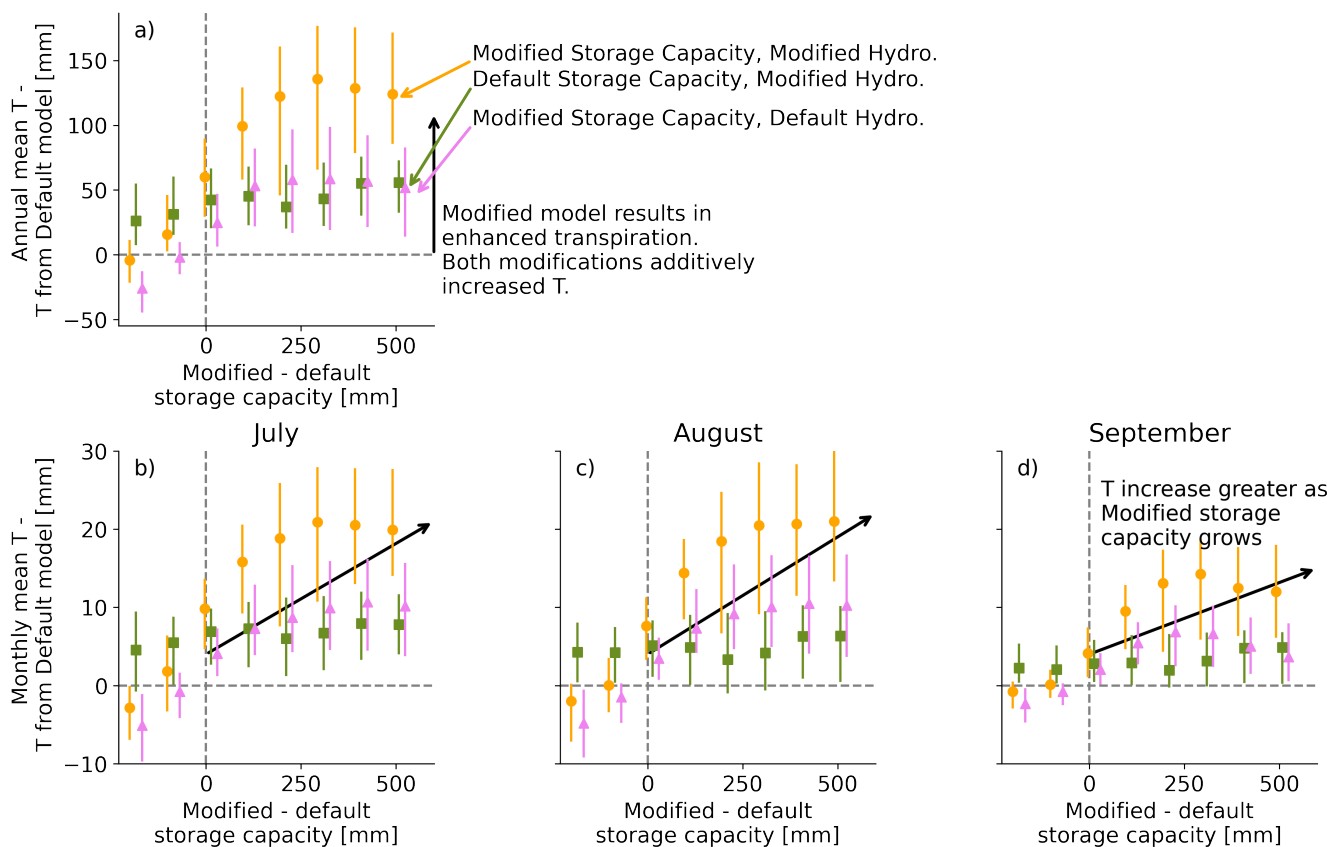

**Figure 8.** Binned histograms of the difference in simulated Transpiration (T) between each simulation and the simulation with default storage capacity and hydrology as a function of the change in storage capacity from the default to modified for (a) annual T and (b)-(d) monthly T for July, August, and September, respectively. Grey dashed lines mark (horizontal) no T change from the default model and (vertical) no storage capacity change between the modified and default storage capacity. Vertical lines for each marked point indicate the spread from 25th-75th percentile. There is little change in T for locations with a change in storage of <100 mm. Greater than 100 mm, though, both the modified hydrology and modified storage capacity separately result in enhanced T, with the combined effect for the fully modified simulation an average of about 100 mm annual increase for sites with large modified storage capacity or a monthly difference of about 20 mm in August. The result is significantly enhanced T, with the largest effect at sites with the largest storage capacity in the modified model.





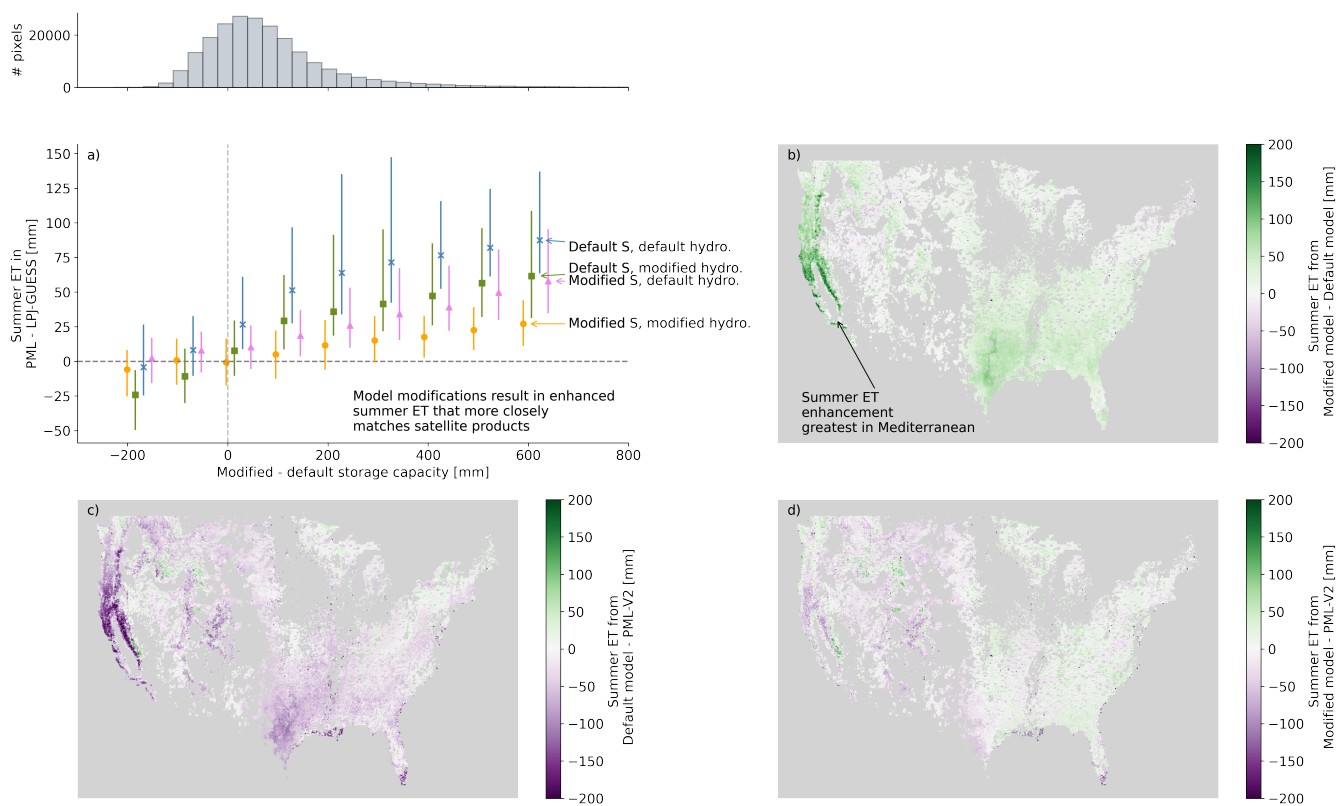

**Figure 9.** (a) Difference in summer ET between each simulation and ET from PML-V2 across CONUS. Points mark median value, and the range shown spans 25-75th percentile. Summer is defined as July-September. As the change in storage capacity grows, the difference in summer ET between PML-V2 and the default model grows as well, with the default model underpredicting ET by an average of about 80 mm. Both modifications separately enhance ET, but the fully modified model most closely matches summer ET from PML. The histogram above shows the distribution of pixels across the storage differences from default. (b) A map of the difference between summer ET in the fully modified model and in the default model. The most important summer gains are along the west coast, particularly in California. Differences between summer ET as modeled by PML-V2 and (c) the default model or (d) the modified model.




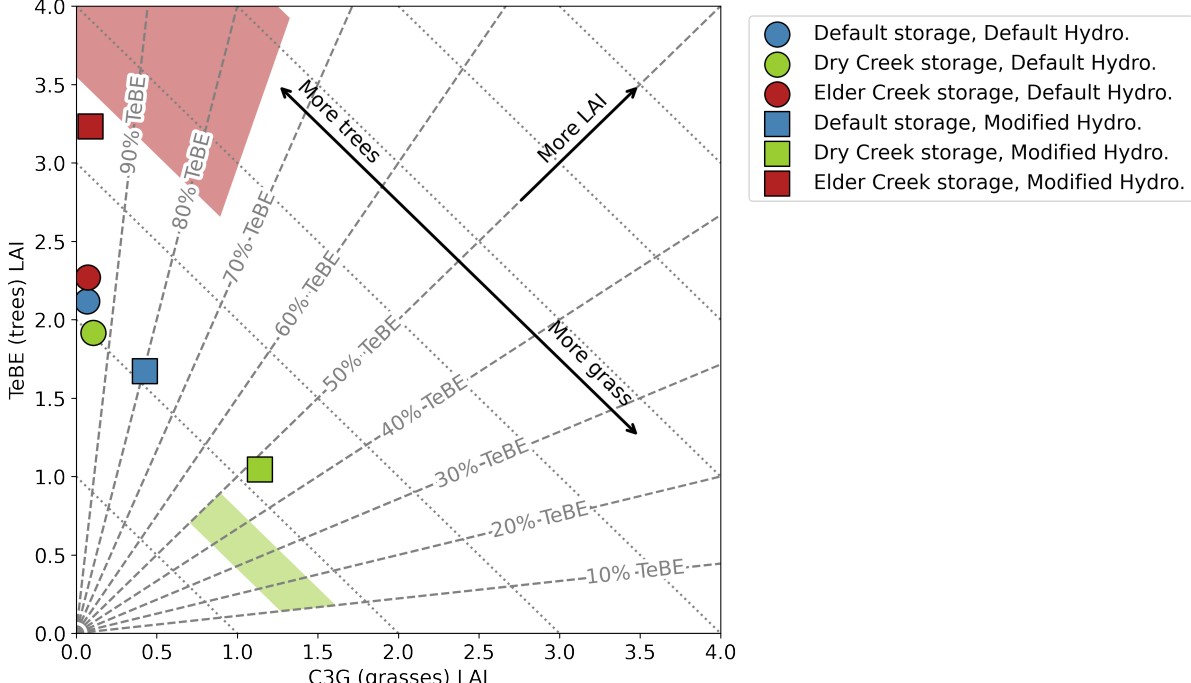

**Figure 10.** Predicted vegetation community, as measured by fraction of LAI devoted to C3G (C3 grass) versus TeBE (temperate broadleaf evergreen tree). The shaded regions denote the actual vegetation community composition at (red) Elder Creek and (green) Dry Creek, estimated from field observations. Dotted lines indicate lines of constant total LAI, and dashed lines indicated lines of constant ratio between TeBE and C3G. Default model configuration (blue circle) does not closely match either the Dry Creek or Elder Creek biome. It is necessary to modify both the hydrology scheme (squares) and the subsurface storage capacity (red for Elder Creek and green for Dry Creek) to achieve the best match to the actual vegetation community.

still very similar to the default storage capacity vegetation community, although Elder Creek has slightly higher LAI than Dry Creek. With the modified storage capacity and hydrology, however, vegetation communities for Elder Creek and Dry Creek are substantially different, with nearly 100% trees at Elder Creek and less than 50% trees at Dry Creek. LAI at Dry Creek is also significantly lower than that at Elder Creek. Neither prediction falls exactly in the shaded region for the site; however these results clearly distinguish the high and low storage sites.

Across CONUS, greater storage capacity is related to an increase in trees (Figure 11a), as was found at the Elder Creek site. Conversely, where storage capacity decreases (as with Dry Creek), the community shifts towards more grass (Figure 11c). Increases in trees are largest in Texas and California (Figure 11b), the same places where enhancements in ET are strongest. Increases in grass are centered in the Great Basin (Figure 11d).



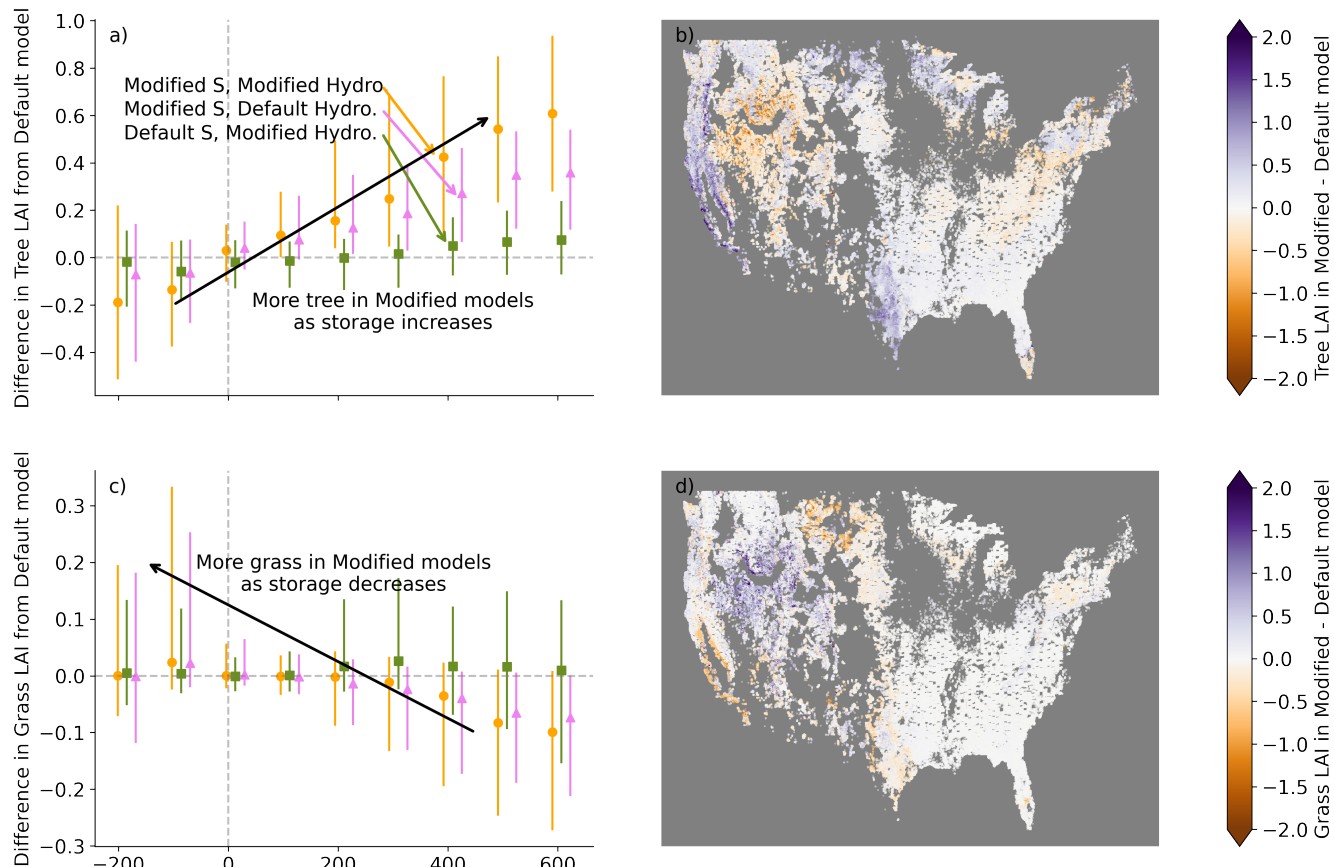

**Figure 11.** Modifications to hydrology and subsurface storage capacity impact the modeled plant community across CONUS. (a) As the modified storage capacity grows relative to the default, more trees are supported. When storage capacity decreases, fewer trees are supported. Conversely, in (b) there is more grass at lower storage capacity and less grass at higher capacity. (b) and (d) show the spatial patterns of changes in vegetation. In (b), more trees are supported along the west coast and in Texas, while fewer trees are supported in the Great Basin. In (c), more grass is supported in the Great Basin. In (a) and (c), points mark median value for each bin, and range spans 25-75th percentile.



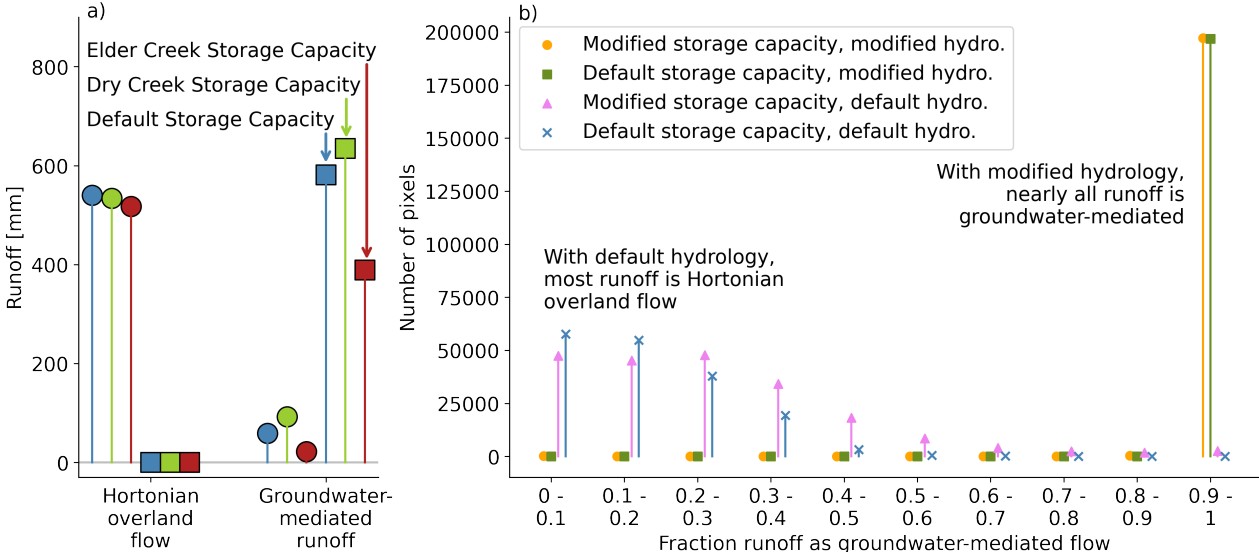

**Figure 12.** Evaluation of runoff partitioning between surface flows and groundwater-mediated flows in the different model setups for (a) the case study sites and (b) across CONUS. With the modified hydrology scheme (squares in panel a, orange circle and green square in panel b), essentially all runoff is groundwater mediated, whereas most runoff is Hortonion overland flow in the default model (circles in panel a, violet triangle and blue x in panel b).

## 3.4 When using modified hydrology, interpretation of dominant runoff generation mechanisms is dramatically different

The modified hydrology scheme results in a massive shift the runoff generation mechanism from mostly Horton overland flow to essentially all groundwater-mediation runoff (Figure 6ac for case study and Figure 6b for CONUS). This change matches our
5  understanding of runoff generation at Elder (Salve et al., 2012) and Dry Creek (Lapides et al., 2022), where Horton overland flow does not occur (Hahm et al., 2019; Dralle et al., 2018). More generally, prevalence of groundwater-mediated runoff matches contemporary understanding of runoff processes in upland, vegetated landscapes.

## 4 Discussion

### 4.1 Where is it most important to account for rock moisture?

10  Across CONUS, the largest enhancements in transpiration (T) with the modified storage and hydrology occurred along the West Coast and in Texas with other visible gains around the southeast (Figure 7a). Both the West Coast and Texas experience long dry periods with substantial precipitation in between (Kottek et al., 2006). However, annual patterns of precipitation delivery are markedly different. On the West Coast, precipitation is delivered primarily in the winter with a predictably dry



summer (Kottek et al., 2006), whereas in Texas, dry periods are scattered throughout the year. Thus, for both areas, the ability to store more water allows for sustained T through dry periods when plants can continue to transpire water stored underground. For the West Coast, the dry period is specifically the summer, so the enhancement in T is particularly clear when examining summer ET only (Figure 9b), whereas the increase in ET is spread throughout the year for Texas and the southeast. Regardless

of the seasonal pattern, enhanced water availability allows for enhanced T and representation of a lusher vegetation community particularly where dry periods are common, and storage capacity is large. The resulting impact on the water cycle can be large (more than 100 mm more T per year, Figure 8a), corresponding to a large change in modeled vegetation community (Figure 11), net primary productivity (Figure S2), and carbon storage (Figure S3).

## 4.2   Further improvements to LPJ-GUESS needed to capture late-summer ET

In the Elder Creek case study, transpiration (T) is extended later into the summer when using the modified storage and hydrology (Figure 6). However, T from LPJ-GUESS still drops before T from PML-V2. This behavior holds for the sites across CONUS with the largest storage, where summer ET is still underestimated by LPJ-GUESS relative to PML-V2 (Figure 9). It is possible that PML-V2 may overestimate the extent to which T continues through the late summer (MODIS may overestimate late summer T; Link et al., 2014). However, if T does continue further into the summer at sites with large storage capacity,

what limits this late-summer T in LPJ-GUESS?

The case study root zone moisture time series provided a clue. For Dry Creek and default storage capacity, all storage water was used up in the summer (Figure 4c). This drop in available storage corresponded to a drop in T. At Elder Creek, T also drops before the end of the summer, but significant water remains in storage (Figure 4c). This is true for minimum storage at large storage-capacity sites across CONUS, where up to a median of about 200 mm of storage remained even at the driest

time of year (Figure 5c). Since simulated transpiration is given by the smaller of water supply or demand, the fact that supply was not used up indicates that the model identified demand-limited (rather than supply-limited) conditions. The limitation on late-summer T was no longer water availability but related to a rate limitation from photosynthetic pathways that are still not fully understood in water-limited conditions (Tezara et al., 1999; Tuzet et al., 2003; Pappas et al., 2013; Zweifel et al., 2006; Vico and Porporato, 2008; Lawlor and Tezara, 2009; Keenan et al., 2010; McDowell, 2011; Tardieu et al., 2011; Sun et al.,

2020). Thus, if it is necessary to further enhance late-summer T for greater model realism, it is necessary to improve the plant physiology in addition to the hydrology scheme and storage to see further gains.

## 4.3   Implications for DGVMs and ESMs

Three key conclusions can be drawn from this work to inform DGVM modelling and Earth system modelling (ESM). Firstly, modeling the ability of plants to access moisture in weathered bedrock has the capacity to improve DGVMs. These improve-

ments will change water fluxes and vegetation cover, and, in the context of simulation with ESMs, also energy fluxes. Including these improvements the land surface components of ESMs has the potential to reduce model biases, particularly in the atmospheric model component which is dependent on the land surface for its lower boundary condition. Given the importance of climate modeling, this would appear to be of high scientific and political priority. However, doing so for global modeling will





be challenging as most regions are relatively data and knowledge poor when compared to the CONUS and the study sites. Implementing weathered bedrock soil moisture in global DGVMs (and therefore ESMs) will likely require a coordinated effort by hydrologists, ecologists, and DGVM and ESM modellers.

Secondly, we note that DGVMs are sensitive to both storage and water flow pathways. Historically, this may not have been fully recognised. Pappas et al. (2013) found that LPJ-GUESS results were sensitive to only one hydrological model parameter: soil storage capacity. Our results corroborate this finding to some extent, but we also found that altering the hydrology scheme was important, in particular setting the maximum rate of infiltration to saturated hydraulic conductivity. However, the increase in infiltration rate with the modified hydrology became important only with large storage capacities (at least 100 mm more than default storage capacity, see Figure 8), which fall outside the range explored by Pappas et al. (2013). Thus, our analysis reveals the importance of improved process representation and well as representing realistic storage capacity.

Finally, improving the simulated hydrology (via, for example, weathered bedrock moisture) will present opportunities to reevaluate and improve representation of related plant processes (in this case plant water demand). This is well evidenced by the case of the simulated late summer T shortfall as discussed above. Furthermore, in some cases some adjustments may even be essential as unrealistic hydrology may have required a compensating error in other processes, and this error will be laid bare under the new hydrology scheme. It might be possible to mitigate such cases with fairly minor adjustment to existing process representations. However, increased realism on the hydrological side presents an opportunity to implement and test alternative approaches for modelling plant responses. In particular, recent developments in understanding plant behaviour using eco-evolutionary optimality approaches (EEO; Stocker et al., 2020; Joshi et al., 2022) may provide alternative process-representations that synergize with improved hydrology to increase overall model realism.

## 4.4 Model simplifications

The modified hydrology and subsurface representations presented in this study are simplified relative to reality. The hydrology scheme is 1-D, and subsurface structure is represented using only two buckets (soil and weathered bedrock). Even with this simple representation, enhanced process realism and realistic subsurface storage capacities significantly improve flux calculations (see Figures 4-9), indicating that previous model limitations were not due primarily to the bucket model structure (as expected from previous studies, e.g., Pappas et al., 2013). However, using only two subsurface layers regardless of the depth does have limitations. For instance, the two-layer model does not allow for plants of different rooting depths within the upper soil layer to have different water access. If the upper soil layer is 150 cm thick, then a grass that roots to 50 cm has the same access to water as a plant that roots to 120 cm–both have full access to the upper soil layer. This setup also provides no ability to prescribe different plant sensitivities to water limitations at detailed sublayers within the root profile. Given the great complexity involved in plant-water uptake, however, and our use of subsurface storage estimates based on actual plant water use (McCormick et al., 2021), it is reasonable to assume that plants have access somehow to all stored water (Feddes et al., 2001). The simplicity of the 2-layer model allows for plants to access all of the available water without prescribing detailed strategies based on root profiles and niches. However, the model setup here does preserve the core logic implicit in the LPJ-GUESS hydrology that grasses have very limited access to deeper soil water, and that trees have more. Further, since layer depths were



determined based on soil properties to achieve the desired storage, layer depths may not correspond to actual depths of water storage in the landscape, so using rooting depths to determine plant water access within the profile is not be appropriate in this model structure.

## 5 Conclusions

In this study, we unite three important themes that have recently emerged from critical zone research efforts, responding to a recent call for action to better incorporate findings from critical zone science into Earth system models (Fan et al., 2019): i) the observation that water sourced from below soil, within weathered bedrock, commonly sustains plant communities through seasonal dry periods (e.g., Rempe and Dietrich, 2018; Rose, 2003; Schwinning, 2010), ii) the observation that the structure—or weathering profile—of this weathered bedrock zone dictates its water storage capacity and seasonal water storage dynamics

(e.g., Dralle et al., 2018; Hahm et al., 2019, 2022), and iii) empirical and theoretical work that suggests that critical zone architecture varies systematically across the landscape as a function of lithology, tectonics, and climate (e.g., Pelletier et al., 2018; Riebe et al., 2017). We bring these insights to bear on global dynamic vegetation models (DGVMs) by incorporating water stored in weathered bedrock into a widely used DGVM, LPJ-GUESS. The addition of this spatially variable deeper moisture store ('rock moisture') along with updates to the hydrology module allowed LPJ-GUESS to capture the differences

in vegetation community and response between two intensively studied sites with similar climate but very distinct vegetation communities. When applied across the contiguous United States, the addition of rock moisture allowed for enhanced evapotranspiration later into the dry season at seasonally dry sites, better capturing observed behavior. This work highlights the importance of accounting for rock moisture in DGVMs and Earth system models and provides a roadmap for the inclusion of rock moisture in other modeling frameworks.

*Code and data availability.* The modified version of LPJ-GUESS DGVM used to produce the results described in this manuscript as well as required input data and produced output will be available on CyVerse at the time of publication.

## Appendix A: Model evaluation–runoff

We evaluated overall model performance by comparing modeled annual runoff, annual ET, and summer ET across CONUS. PML-V2 (Zhang et al., 2019) was used as the reference ET dataset, and a reference runoff dataset was calculated as the

difference between precipitation from PRISM (PRISM Climate Group, 2014) and ET from PML-V2. We compared mean annual runoff and mean summer ET at each pixel (Figure A1 for runoff and Figure 9 for summer ET). There is essentially no difference in performance at estimating annual runoff among the different LPJ-GUESS model scenarios tested (Figure A1. However, summer ET model performance for pixels with a large subsurface storage capacity was significantly improved with the LPJ-GUESS model with modified storage and hydrology (Figure 9).



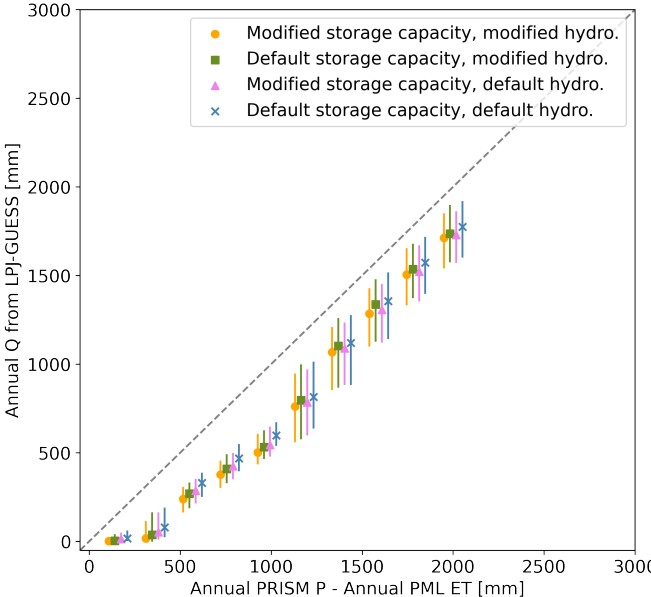

**Figure A1.** Mass balance check showing that the model modifications do not affect mass balance. Runoff from LPJ-GUESS is slightly lower than that calculated via mass balance from PRISM precipitation and PML-V2 ET across all pixels and simulations.

We further explored model performance on runoff and ET using Kling-Gupta Efficiency (Gupta et al., 2009) and a metric determining effectiveness at capturing spatial distribution, calculated following Seiler et al. (2022) as:

$$S_{dist} = 2(1 + R)(\sigma + \frac{1}{\sigma})^{-2}, \tag{A1}$$

where $R$ is the correlation coefficient between a mean annual variable from an LPJ-GUESS model scenario and a reference
5   data set, and $\sigma$ is given as:

$$\sigma = \frac{\sigma_{model}}{\sigma_{reference}}, \tag{A2}$$

where $\sigma_{model}$ and $\sigma_{reference}$ are the standard deviation of a mean annual variable for LPJ-GUESS model scenario and the reference data set, respectively.

For annual runoff and annual ET, KGE with the reference data decreases from the standard model to the modified model.
10   The difference is modest but measurable in both cases. The spatial distribution of annual runoff and ET are captured essentially the same by all model scenarios. Dry season ET performance, however, is substantially improved from the standard to the modified model from a KGE of 0.27 (poor performance) to 0.89 (excellent performance). The spatial distribution is also captured significantly better (improves from 0.66 to 0.96).



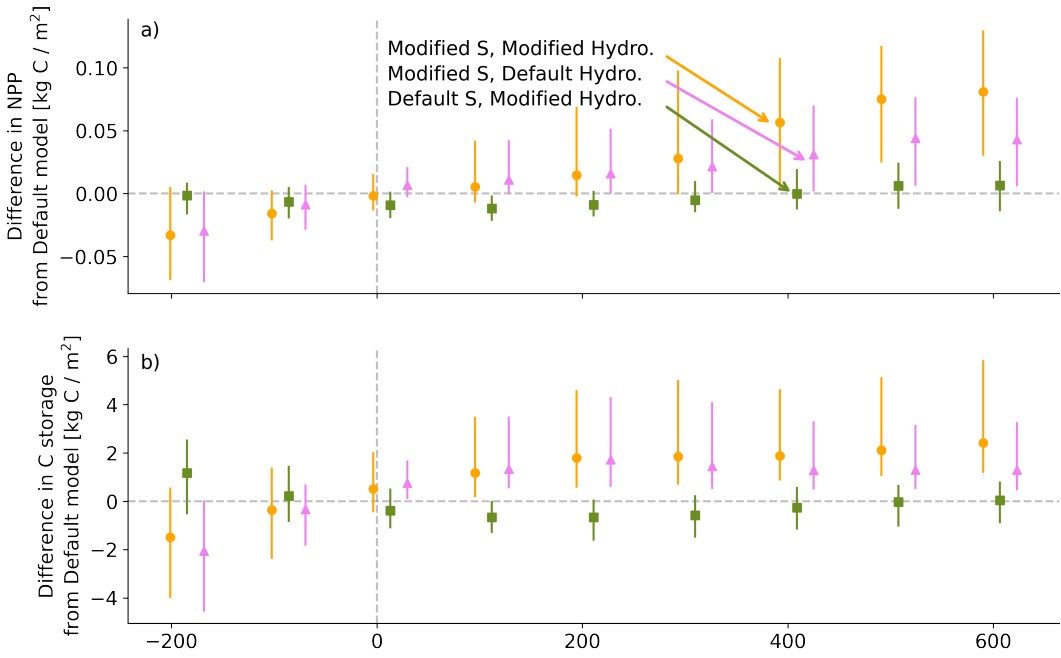

**Figure A2.** Impact of modified hydrology and storage relative to default LPJ-GUESS model on (a) Net primary productivity (NPP) and (b) Carbon sequestration. Both NPP and C are enhanced with modified storage and hydrology schemes.

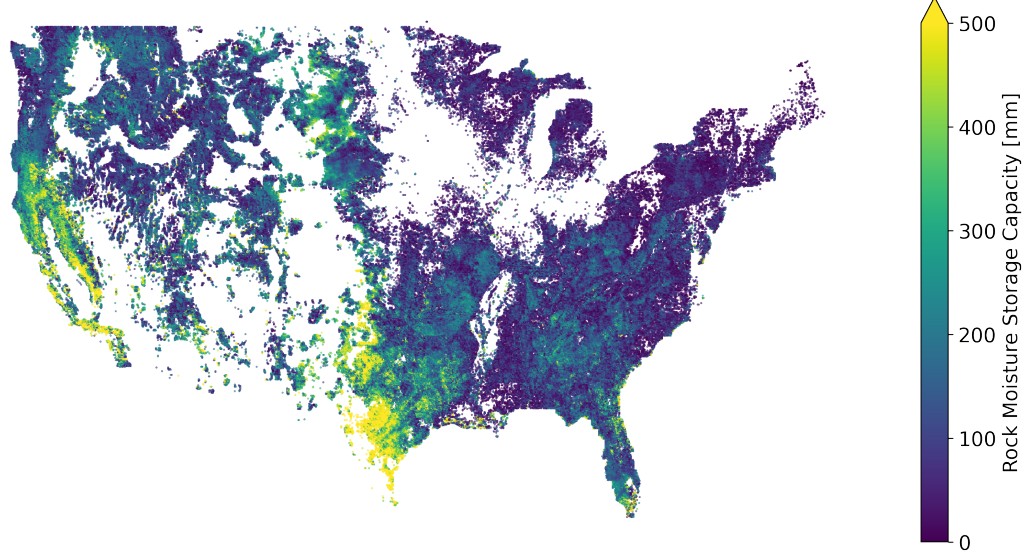

**Figure A3.** Rock moisture storage across CONUS, as described in Section 2.3 of the main text. This storage reservoir represents water help below soils in weathered bedrock and is given by the difference between total estimated root zone storage capacity and estimated soil storage capacity.



| Hydrology | Storage capacity | Runoff KGE | Runoff $S_{dist}$ | ET KGE | ET $S_{dist}$ | ET KGE (dry season) | ET $S_{dist}$ (dry season) |
|---|---|---|---|---|---|---|---|
| Standard | Standard | 0.57 | 0.96 | 0.90 | 0.95 | 0.27 | 0.66 |
| Standard | Modified | 0.51 | 0.96 | 0.82 | 0.94 | 0.81 | 0.92 |
| Modified | Standard | 0.49 | 0.94 | 0.80 | 0.93 | 0.54 | 0.78 |
| Modified | Modified | 0.45 | 0.96 | 0.73 | 0.93 | 0.89 | 0.96 |

**Table A1.** Caption

**A1**

*Author contributions.* WJH, MF, TH, and DND conceived of this study. DAL, WJH, and DND developed modified model code. DAL ran model simulations and produced publication figures. All authors contributed to writing and editing the manuscript.

*Competing interests.* No competing interests are present.

5    *Acknowledgements.* We would like to acknowledge the US-NSF CZO SAVI International Scholars Program, Simon Fraser University, Natural Sciences and Engineering Research Council of Canada – Discovery Grant, and the Canadian Foundation for Innovation - British Columbia Knowledge Development Fund JELF Grant for supporting this research. This research used resources provided by the SCINet project and the AI Center of Excellence of the USDA Agricultural Research Service, ARS project number 0500-00093-001-00-D.



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
