# Peer review of "Inclusion of bedrock vadose zone in dynamic global vegetation models is key for simulating vegetation structure and functioning"

_EGUsphere, 2023_

## Author Comment (AC1)

**Reviewer #2:**

Reviewer summary:

The study of Lapides et al. investigates the inclusion of rock moisture into a global dynamic vegetation model. To test the approach the team tests modifications to the storage compartments of the model and compare the model results to two study sites and available data for the continental US.

Overall the authors present a very strong research paper, which unfortunately currently lacks clarity and a discussion of uncertainty. Due to imprecise language and a lot of figures it is sometimes hard to follow the authors in their conclusions. Moving 1-3 figures to a supplement or appendix would help to tell a clearer story.

Thank you for this feedback. We will respond (in blue) below to these comments (in black) as the reviewer expands upon them.

A lot of figures spaced out over short and precise explanations. Moving ~3 Figures to the supplement would help clarity a lot. Currently very hard to read. Figure 12 seems not to be referenced and explained anywhere.

We received similar feedback from Reviewer #1 in regard to the number of figures. Following Reviewer #1's suggestions, we have moved two figures from the main text to the supplement and simplified the remaining figures (see response to Reviewer #1). As for Figure 12, we included the wrong reference in Section 3.4. We have updated the previously incorrect citation to Figure 7 to Figure 12.

I would also appreciate if the authors could comment on how the data uncertainty influences the results. As the authors note data in the US is relatively good compared to global soil and bedrock datasets. It would greatly improve the study if the authors could provide an insight of how their results are impacted by data uncertainty in the presented study regions but also what this possibly entails for the global scale

Thank you for this suggestion. We added a paragraph in the discussion section addressing uncertainty and extending datasets globally:

"The data sources used in this study for soil capacity and to calculate weathered bedrock storage capacity are specific to the United States. To extend this model beyond the United States, it would be most important to extend estimates of total plant-accessible storage since

soils datasets are generally inadequate to represent plant-accessible water stores even where they are available (McCormick et al., 2021), due to widespread plant water uptake from layers deeper than those in traditionally mapped soils databases. While the specific distributed water flux datasets used in this study are not globally available, Wang-Erlandsson et al. (2016) used a similar deficit-based strategy to the one used in this study to estimate plant-accessible water storage globally with alternative water flux datasets. The accuracy of estimates of this type is limited by both (i) the accuracy of the input water flux data and (ii) the time period of data availability. In the case of the present study, the PML-V2 evapotranspiration and PRISM precipitation datasets used to calculate the root-zone storage deficit close mass balance well with USGS streamflow gages (Rempe et al., 202x) in undisturbed watersheds in the western United States. However, data concordance should be confirmed with any data sources to be combined for use in a root-zone storage deficit calculation since mass balance errors can compound over time. Second, the time period of data availability is important since the maximum root-zone storage deficit provides only a minimum bound on plant-accessible water storage. With a longer timeseries, the minimum bound is more likely to approach the actual plant-accessible water storage, particularly if dry periods or disturbances like fire or logging are included in the timeseries. Shorter timeseries or timeseries that fall during a particularly wet period of history may be more likely to underestimate plant-available water storage. Based on the findings of this study, underestimating storage capacity would result in lower evapotranspiration and less tree growth in LPJ-GUESS, and overestimating storage capacity would result in the opposite. "

Linked to the comment on uncertainty the authors should provide a clearer link to how the two sites are representative for the US and globally. What other regions should future research investigate to figure out on how to represent rock water on a global scale in these models?

We have added a paragraph to the discussion in Section 4.2 on this topic:

"Elder Creek (480 mm storage capacity) and Dry Creek (180 mm storage capacity) have storage capacities at the 79th and 4th percentiles of storage capacities in the Mediterranean region included in this study (Cs label in Figure 8e). As such, they capture two broad sets of behavior found in Mediterranean sites that are also common beyond Mediterranean regions, but they do not fall at the mode of the distribution of storage capacities, which is 330 mm. Thus, it would be valuable to continue with more site-specific studies to identify whether additional complexity or alteration to the model structure would be valuable. In particular, it would be valuable to explore rock moisture dynamics in DGVMs in snow-dominated sites, which was not explored in detail in this study."

The authors now have modified mainly the storage component of the model. Would it be beneficial to also provide a lateral groundwater connection inside the model?

Absolutely. Fan et al. (2019) advocate for landscape position as an important driver of rooting depths and therefore vegetation dynamics. However, incorporating lateral groundwater flows into LPJ-GUESS would require fundamental changes to the model structure that are far beyond the scope of this study. Given the extensive changes this would require, it may make sense to explore this question by coupling a DGVM with a more complex hydrological model (like the coupling of CLM and LPJ for CLM-DGVM). However, this is still a good point, and we added a note to the discussion to address this:

"Other aspects of hydrology may also be essential to account for in certain regions or landscape positions, such as lateral groundwater flows (Fan et al., 2019). However, most DGVMs are not structured to account for topography, making the inclusion of both subsurface and surface water flow subsidies highly challenging. Future efforts could explore more complex hydrology by restructuring a DGVM like LPJ-GUESS to take into account topography or coupling the plant dynamics in a DGVM such as LPJ to an existing hydrological model."

Fan, Ying, et al. "Hillslope hydrology in global change research and earth system modeling." *Water Resources Research* 55.2 (2019): 1737-1772.

Additional detailed comments:

P2 11: You motivate your paper with the Mediterranean and then evaluate it for the US, why?

We think that the reviewer's question may relate to the two possible definitions of "Mediterranean regions," which we did not adequately define in the original manuscript. First off, we define Mediterranean areas as those with a Mediterranean climate. We adjust the language throughout to "regions with Mediterranean climate" and include a brief description of a the Mediterranean climate: "which experience hot dry summers and cooler, wetter winters." We also added a sentence clarifying the importance of carryover moisture in mediterranean regions on page 3, line 2:

"Generally, it is the case that water stored from the wet winter is essential for supporting plant function during the dry summer in regions with Mediterranean climates."

To further address this comment: we motivate the study with areas with Mediterranean climate, which provided a clue that something may be missing from the models. We then want to evaluate it at a large scale, not just in areas with Mediterranean climate, to demonstrate that the model change is generally applicable, not something that applies specifically to areas with Mediterranean climate. Areas with Mediterranean climate are the most affected, but models can be generally improved

everywhere if the subsurface hydrology and structure is properly represented. We added notes to that effect at revised page 2 lines 11 and page 4 line 14:

"This performance gap provides a clue that there may be an essential component missing from these models. "

"We test this hypothesis in detail at two intensively monitored sites in Northern California with similar climate but distinct vegetation communities (Hahm et al., 2019) and more broadly at 4 km resolution across CONUS to demonstrate that these changes result in realistic predictions not just in Mediterranean areas but across all biomes represented in CONUS."

P3 1: Again unclear.

It wasn't clear to us what the confusion was. Could you be more specific so we can improve the text?

P4 10: You lost me. How do these places relate to the MED issues you highlighted? How much will they be transferable?

We understand the confusion. We think that we may have not explained clearly enough what the change is. We are not just adding bedrock water underneath the existing LPJ soil storage but altering the total available storage in the subsurface layers, as described in the methods. While this is clear in the methods, it is not clear in the introduction. To help with this issue, we added a statement at lines XX:

"We expect to see the largest impacts in Mediterranean areas, but these improvements should show up more modestly in other areas as well since this change will result in a more realistic depiction of subsurface water availability everywhere."

Table 2: Nice but very difficult to read. Could you move the justification just to text and pivot the table?

Great suggestion. We pivoted the table, and left the justification to the text.

P8 1-4: This is unnecessary. Cite one key paper and be done with it. This seems more like self-advertisement than actual scientific proof

We shortened the list of references.

P8 6: More recent versions than what? The one used in this paper? If so this doesn't matter then. Or if it matters explain why.

Yes, we are referring to versions more recent than included in this paper. We have removed the comment.

P11 23: In the model or in the real-world site?

Both! We clarified with the following comment:

"For the case study locations both in reality and in the model..."

P11 27: For what time frame?

This is clearer now that we have added more details about the model run in response to Reviewer #1, but we also added a comment at Line XX for further clarity:

"for the full study period (1981-2021)"

P11 28: First reference to PML-V2 is that a dataset or a model? Why do you use it as benchmark?

We added a clarification on what PML-V2 is ("distributed ET data product PML-V2"). PML-V2 is first mentioned in the Section 2.3 (prior to this), in which we note that "PRISM precipitation and PML-V2 have been found to perform well for mass balance closure compared with USGS streamflow gages (Rempe et al., 20xx)."

29: Why did you not use a hydrological model which might compute runoff instead?

Using a hydrological model would introduce additional uncertainty and complexity into the study. For simplicity and for consistency with the data used for evaluation in this study, the mass balance approach provides a simple estimate of runoff that incorporates no lateral water flows (similar to LPJ-GUESS), making it a good comparison.

P12 8: Unclear and confusing sentence. The mass balance of what? What spatial distribution metric and what is it used for?

We updated that sentence for clarity. It now reads: "We used annual runoff from mass balance between PRISM P and PMl-V2 ET for comparison with LPJ-GUESS runoff and annual and summer ET from PML-V2 for comparison with LPJ-GUESS ET to evaluate model performance based on Kling-Gupta Efficiency"

10: This is a result and belongs into section 3.

We moved this information to Section 3.3 in the results: "In terms of overall model performance, summer ET improvements with the fully modified model drive strong model improvements, although annual runoff and overall ET performance is slightly decreased (Supplemental Information S1)."

Fig 3: Missing section reference. This does not need to be an extra figure, e.g., add as small legend to Fig. 5.

We tried to turn Figure 3 into an inset. However, the descriptive text, which we think is very useful, becomes hard to read, so we left Figure 3 as a separate figure to improve readability of the paper.

Fig 5: Please explain first what a-d show. It is unclear whether a and c both show results for the whole CONUS. It is currently easy to get lost in the information.

We removed extra information from the caption and clarified that panels a, c, and d show results across CONUS, while b shows only pixels across CONUS where storage increases by at least 200 mm.

P15 18: I assume T stands for transpiration? Make it explicit also in Figure 6

We clarified that T stands for transpiration at both of these locations.

Fig. 6: The effect seems strongest if storage capacity is large but what is the explanation for transpiration underestimation in small capacity sites?

We agree that there is still a lingering question here. This is addressed in the discussion in Section 4.2.

PML should be a different color since you already use black to indicate change. Also, the colored arrows make the plot hard to read and may be confused with datapoints. For all figs: the annotation is helpful but it should be very clear that it is an annotation. Should also be added to legend in all plots.

We changed the color for PML to grey and added legends instead of the labeled lines so that all annotations are clear.

Fig 7 and 8: Maybe these two could be combined? Because they do not show that much new content and you already have a lot of complex figures and in the text and you jump between these two a lot. Maybe some of this could be moved to the supplement?

At the suggestion of Reviewer #1, we moved Figure 8 to the supplement. We agree that there is a lot of similar material.

P16 6: Indeed, but where are the reductions the lowest and why?

This is a good question. Comparing Figure 9c to 9d, we see that error is reduced essentially everywhere, and the error becomes centered around 0 rather than mostly negative. Where error reductions are small, error was small to begin with. As demonstrated in Figure 9a, sites with very little difference between the modified and default storage capacity or with a smaller storage capacity in the modified model tend to see little change in the resulting summer ET signal. We added a comment about this to Section 3.2: "The change in storage capacity between the modified and default models does an excellent job determining how large of a change there will be to summer ET (Figure 9a), so that places with little change in summer ET are those where the storage change was negative or very small."

P22 4: Is this supposed to be Fig 12?

Yes, thank you. We have corrected that.

---

## Author Comment (AC2)

Reviewer 1:

Thank you so much for the detailed and thoughtful feedback. We appreciate that you took the time to engage with our work. We have responded (in blue) to the reviewer comments (in black) as numbered below (and re-organized the order such that comments with responses come after general reviewer summary comments).

Reviewer Summary Comments:

1. The contributions of subsoil layers to the water supply of forests has been documented since the mid-1990s. What was originally interpreted as a rare and unusual phenomenon is now recognized as essential to the ecohydrology of many terrestrial biomes. There have been calls to implement a broader definition of the rhizosphere in global vegetation models to include subsoil strata. Apart from Jimenez-Rodriguez et al (2022) cited by the authors, the present manuscript may be the only other attempt towards this goal. Thus, the paper is timely and important.

2. The authors present a novel tool for predicting the influence of plant-available subsoil water on vegetation composition and the hydrological cycle. This involves a restructuring of the hydrological scheme in the DGVM LPJ-GUESS, an age-structured plant functional type model constructed to predict global biome distributions through an optimization process that maximized NPP. The modification was introduced as having two independent components (Fig. 3): the increase in storage by incorporating the subsoil and the increase in subsoil recharge by modifying runoff prediction. As it turns out, are both needed for the best fit.

3. The substantial result is that root access to subsoil ('rock moisture') AND greater partitioning of precipitation into the subsoil are essential to more accurately predicting tree LAI and summer ET across the continental United States (and in the case study).

7. The authors do a good job presenting their work in the context of prior contributions.

8. The title is fine.

9. The abstract is also fine.

11. The language is fine.

12. The symbols are consistently used and have correct units.

14. The references are appropriate.

4. The manuscript is very clear explaining the hydrological modification that affects runoff generation (Q_surf v Q_baseflow). However, I missed an equally clear explanation of the way in which root uptake of water is regulated in the model and how that scheme was adapted to the restructuring of subsurface storage. If that part of the LPJ-GUESS hasn't changed from earlier versions, (Haxeltine & Prentice 1996; BIOME3), water supply is downregulated through a simple linear correlation with the amount of plant-available soil moisture remaining in two soil layers. Furthermore, tree and grass PFTs are distinguished by their relative access to these pools (e.g., grasses = 90% roots in the to 50 cm, versus trees 33%). I would urge the authors to expand on this aspect of the model description. Note that on page 24 lines 25-29 they are actually addressing the issue of root distribution, so it only makes sense to talk about this up-front. I would be curious to know if grasses had access to the second layer (the subsoil) per default parameters.... Seems that they do per line 32 on the same page. The BIOME3 model might not assign *detailed* strategies on root profiles, but it does have a simple and essential one when it comes to distinguishing trees and grasses competing for two soil water pools. One has to look and see if that makes still sense after the redefinition of the two pools.

We agree that the root water uptake was not adequately explained. We focused on the changes we made to the model rather than the overall functioning, but the root water uptake is essential to understanding the changes made to the model. Furthermore, our choice of water uptake scheme within LPJ-GUESS rendered the values of root distributions for trees unimportant. The grasses do, as the reviewer mentioned, still have 90% of their roots in the top 50cm. Unfortunately, we neglected to mention and describe this, apologies for this oversight. We believe that this water uptake scheme gives a better reflection of reality, because trees can essentially explore the whole volume (reflecting that in reality they can utilize tap/coarse roots), whereas grasses are limited to 90% in the top 50cm (no taproots). We have added a description of the existing water uptake strategy as implemented in LPJ-Guess to the methods as follows:

"We used the so-called ``SMART" root water uptake scheme implemented in LPJ-GUESS. This maintains a key feature of the current default water uptake scheme that the supply of water for transpiration is not curtailed until soil water content reaches wilting point (which stands in contrast to previous versions of LPJ and its ancestor BIOME models). In the SMART scheme, unlike the default water uptake scheme, trees are not constrained to access water according to prescribed root distributions. By removing this constraint on trees, we believe that the SMART

scheme better reflects the ability of trees to forage for water throughout the available subsurface storage volume using their taproot and other coarse roots. This is supported by our finding that the SMART water uptake strategy allows transpiration to continue further into the summer (more closely matching real transpiration patterns) than any other root water uptake model implemented in LPJ-GUESS (Supplemental Figure A6). This also is aligned theoretically with our approach for determining the subsurface storage capacity, which is sized to hold all of the water that plants are known to have access to. As such, trees should be able to access all of the water stored in the subsurface in either layer. Furthermore, model parsimony is improved by effectively removing the rooting depth parameters. This has the further benefit of avoiding the necessity to reconcile rooting depth profiles developed for the fixed soil layer depths in the default LPJ-GUESS model with the new subsurface structure with spatially variable layer depths. Grasses, however, follow the default root uptake behavior in which they have 90% of their roots in the upper soil layer, with only 10\% of their roots in the lower layer. Their maximum water uptake rates are weighted by this rooting profile regardless of layer depths, implying that grasses have limited access to the lower soil/weathered bedrock water pool and can draw a maximum of 10\% of their water from it. Again we believe this is a reasonable representation of reality because, without coarse roots, grasses mostly draw water from near the surface but may be able to root deeper to some extent if needed."

5. Related to the omission of addressing plant interactions with water storage pools, page 23, lines 16 – 26 was a bit undeveloped. Seems to me, one should always be able to find out why a model acts in a certain way. Furthermore, I don't see why residual storage water at the end of summer is necessarily a problem. One would in reality not expect all water to be used by the end of summer every summer.

The continued downregulation of transpiration late in the summer remains interesting and did not receive enough attention. We agree that it is not necessarily a problem for water to remain in storage at the end of the summer. This indicates that the limitation on transpiration is not entirely water availability (because water was available but not used) but something else related to the plant processes (we clarified this in the discussion). However, this does merit more discussion that was in the original draft. We performed additional analyses, as described below.

From page 11, line 1, I gather that the simulations were run for the period from 1981-2021, and so I assume that model output is composed of multi-year averages, suggesting that on average there should be positive residual moisture.

Yes, additional details on the model runs are now added to the methods section:

"For all model runs, the nitrogen cycle was enabled, and land use was not included, so simulation results represent potential natural vegetation. For all locations, we ran four different simulations based on the same climate data for the period 1981-2021 using a 500-yr spin up

period. Results are shown as a mean over the period 1981-2021."

Furthermore, given that Fig. 4 was a model prediction, the fact that 'rate limitation from photosynthetic pathways are still not fully understood' was sort of beside the point. Perhaps the point can be made in relation to Figure 6, though.

We also agree that the comment about photosynthetic pathways was not well-contextualized, and we have clarified (see revised text copied below) that the comment is meant to indicate that the state of knowledge about uptake down-regulation in water-limited conditions may limit our ability to model these processes:

"Since simulated transpiration is given by the smaller of water supply or demand, the fact that supply was not used up indicates that the model identified demand-limited (rather than supply-limited) conditions. The limitation on late-summer T was no longer water availability but related to a rate limitation from photosynthetic pathways that are still not fully understood in water-limited conditions \citep{tezara1999water,tuzet2003coupled,pappas2013sensitivity,zweifel2006intra,vico2008modelling,lawlor2009causes,keenan2010soil,mcdowell2011mechanisms,tardieu2011water,sun2020response}. Thus, if it is necessary to further enhance late-summer T for greater model realism, it is necessary to improve the plant physiology in addition to the hydrology scheme and storage to see further gains. "

However, I am not convinced that the problem lies necessarily in the physiology: It is very possible that assumed water and root distribution could affect the calculation of optimal LAIs in the LPJ-GUESS model. Really, the model probably needs three layers to maintain the partitioning of soil moisture between grasses and trees. At least for the case study, the authors could have tried to optimize the assumed root distributions, or at least do a sensitivity analysis to investigate the question of root distribution as another source of uncertainty. In my opinion this would add a useful message, rather than dismissing the topic out of hand.

Thank you for this suggestion. In our original exploration, we did adjust some parameters in the model to see if the late-summer transpiration would bump up (for instance, the emax parameter that sets the maximum transpiration allowed on a given day), but we did not see significant changes. We have now tried adjusting the root distribution and switching the water uptake routine as well, and these results have been added to the supplement and referenced in the discussion:

"This observation indicates that the limitation on late-summer T was no longer water availability but something related to plant physiology or root water uptake. To rule out root distribution or water uptake strategy, we perturbed the root distributions (10\%, 40\%, and 90\% of roots in

upper soil layer) and applied three of the built-in water uptake schemes (smart--used in this study--, root distribution-based, and water content-based) in the Elder Creek case study site. Across all of these permutations, none resulted in an enhanced transpiration signal that extends later into the dry season than the results presented in the main text (see Appendix C), indicating that plant physiology routines are driving the down-regulation of T late in the summer."
Here is the new figure in Appendix C that demonstrates this:

[Figure]

6. It would have been helpful to have more information on the implementation of the model. From page 11, line 1, I gather that the simulations were run for the period from 1981-2021, and so I assume that model output is composed of multi-year averages and that tree and grass LAIs were optimized over the same period. But this has not been explicitly stated in the manuscript.

Thank you for pointing out the need for further detail on the model runs. We have added a statement with more information:

"For all locations, we ran four different simulations based on the same climate data for the period 1981-2021 using a 500-yr spin up period. Results are shown as a mean over the period 1981-2021."

10. In general, it would have been better to have fewer and/or less complex figures. The figure content was excessively comprehensive, given that the main results were quite straight forward. For example, after the first few results, it is quite evident that the second storage pool needs the enhanced recharge to have the desired effect on ET. Once this is established (and the most fitting place to establish this in the case study, e.g. Fig. 10 is really good in this respect), it is perhaps enough to contrast only the

default model, the fully modified model and the ET data product. Perhaps consider Figs 5 and 8 for supplementary data.

Thank you for the suggestions. We have moved Figures 5 and 8 to supplemental material. We have also reorganized the results so that (the original) Figure 10 is used to establish that we need to compare only the default and fully modified models in the remainder of the study. We then removed the partially modified models from all additional figures.

13. See comments above: more should be said about the way in which LPJ-GUESS predicts functional type composition and how plants interact with storage, i.e., how transpiration is constrained by supply not demand.

We have (described above) added description of the root water uptake strategies. We also added a methods section describing briefly how LPJ-GUESS determines PFT:

"LPJ-GUESS is a dynamic global vegetation model, which simulates how different Plant Functional Types (PFTs) compete for resources (here light, water and nitrogen). The traits of the PFTs determine which PFTs are most successful and thus reach the largest biomass or cover under given environmental conditions. For example, a summer- or raingreen phenology is beneficial in seasonal environments, and PFTs with such a phenology then outcompete evergreen PFTs because individuals grow faster. Root distributions influence the competition for water, whereby deep rooting yields more water access in Mediterranean areas with winter rain. These outcomes are not predefined but they emerge from the functional traits of the PFTs in a given environment. The distribution of PFTs is further constrained by bioclimatic limits (adopted from Sitch et al. 2003) and disturbance by wildfires also affects vegetation dynamics"

15. I recommend expanding the supplementary information section, in exchange for striving for greater synthesis in the results figures.

We have taken this advice, as described above in the responses to comments 5 and 10.

**Reviewer #2:**